# Ultrastructural visualization of 3D chromatin folding using volume electron microscopy and DNA in situ hybridization

Paweł Trzaskoma[1,8], Błażej Ruszczycki[1,8], Byoungkoo Lee [2,8], Katarzyna K. Pels[1], Katarzyna Krawczyk[1], Grzegorz Bokota [3], Andrzej A. Szczepankiewicz [1], Jesse Aaron [4], Agnieszka Walczak[1,5], Małgorzata A. Śliwińska [1], Adriana Magalska[1], Michal Kadlof[3], Artur Wolny[1], Zofia Parteka [3], Sebastian Arabasz[6], Magdalena Kiss-Arabasz[6], Dariusz Plewczyński [3,7], Yijun Ruan[2✉] & Grzegorz M. Wilczyński[1✉]

The human genome is extensively folded into 3-dimensional organization. However, the detailed 3D chromatin folding structures have not been fully visualized due to the lack of robust and ultra-resolution imaging capability. Here, we report the development of an electron microscopy method that combines serial block-face scanning electron microscopy with in situ hybridization (3D-EMISH) to visualize 3D chromatin folding at targeted genomic regions with ultra-resolution (5 × 5 × 30 nm in xyz dimensions) that is superior to the current super-resolution by fluorescence light microscopy. We apply 3D-EMISH to human lymphoblastoid cells at a 1.7 Mb segment of the genome and visualize a large number of distinctive 3D chromatin folding structures in ultra-resolution. We further quantitatively characterize the reconstituted chromatin folding structures by identifying sub-domains, and uncover a high level heterogeneity of chromatin folding ultrastructures in individual nuclei, suggestive of extensive dynamic fluidity in 3D chromatin states.

[1] Nencki Institute of Experimental Biology, Polish Academy of Sciences, 3 Pasteura St, 02-093 Warsaw, Poland. [2] The Jackson Laboratory for Genomic Medicine, 10 Discovery Dr, Farmington, CT 06032, USA. [3] Center of New Technologies, University of Warsaw, 2c Banacha St, 02-097 Warsaw, Poland. [4] Janelia Research Campus, Howard Hughes Medical Institute, 19700 Helix Dr, Ashburn, VA 20147, USA. [5] Department of Gene Expression, Institute of Molecular Biology and Biotechnology, Faculty of Biology, Adam Mickiewicz University, 6 Uniwersytetu Poznanskiego St, 61-614 Poznan, Poland. [6] Łukasiewicz Research NETWORK – PORT Polish Center for Technology Development, 147 Stablowicka St, 54-066 Wroclaw, Poland. [7] Mathematics and Information Science, Warsaw Technical University, 75 Koszykowa St, 00-662 Warsaw, Poland. [8]These authors contributed equally: Paweł Trzaskoma, Błażej Ruszczycki, Byoungkoo Lee. ✉email: Yijun.Ruan@jax.org; g.wilczynski@nencki.gov.pl

How the two-meter-long human genome is folded into the micrometer-sized nuclear space is an important question in biology. Indeed, specific three-dimensional (3D) organization of chromosomal folding has been shown to have a profound impact on genome functions, such as gene transcription[1] and DNA replication[2]. To investigate the detailed structures of 3D genome organization, two general approaches have been developed and applied—sequencing-based mapping and microscopy-based imaging. Mainstream 3D genome mapping methods such as Hi-C[3,4] and ChIA-PET[5] detect pairwise long-range chromatin interactions by chromatin proximity ligation followed by high-throughput sequencing and mapping to the reference genome, thus inferring long-range chromatin contacts and rendering the 3D configuration of the genome. However, 3D genome mapping data are derived from millions of cells[6] and provided an average view of genome folding. Specifically, the techniques revealed smaller subunits of chromosome territories—topologically associated domains (TADs by Hi-C)[7–9] and chromatin contact domains (CCDs by ChIA-PET)[10], demarcated by multiple binding sites for CTCF protein[11]. However, even though the contact probabilities are clearly displayed, the mapping data lack physical scale (e.g., in micrometers and nanometers). To overcome these limitations, microscopy can be used to visualize the actual genome architecture and metric scale in various spatiotemporal resolutions in individual nuclei with different DNA staining methods. Standard fluorescence light microscopy has an optical resolution of about 250 nm[12] and has been applied to image mammalian nuclei, successfully establishing the concept of chromosome territory[13] showing chromosomal morphologies in different cell cycle phases, and is also routinely used in cytogenetics to study abnormal chromosomes[14]. Recently, a super-resolution fluorescence light microscopy ($20 \times 20$ nm in $xy$ dimensions, 50 nm in $z$ dimension) with sequence-specific DNA-binding probes was applied to visualize specific chromatin folding structures for a 10–500 Kb and 3 Mb target genomic region in Drosophila cells[15,16], and 1.2–2.5 Mb target genomic region in human cells[17]. However, this method has a limit in $z$-dimension image depth (up to 3 μm)[17], and thus larger 3D chromatin structures beyond this limit would be truncated or lost. Electron microscopic in situ hybridization (EMISH) using biotin-labeled DNA probes coupled with diaminobenzidine staining has been used to image chromosomal DNA in the nuclei[18]. When standard FISH protocols are used in electron microscopy (EM) study, severe artefacts have been attributed to formamide and high-temperature treatment[19]. However, volume (3D) EM approaches —serial block-face scanning electron microscopy (SBF-SEM)[20,21] and focused ion beam scanning electron microscopy (FIB-SEM)[22] were used with success in studies of global architecture of cell nuclei. SBF-SEM can reach a resolution $\sim 5 \times 5$ nm in $xy$, while resolution in $z$ depends on a thickness of cut ultrathin slices, and FIB-SEM achieves a resolution down to $\sim 3$ nm in all dimensions[23]. The most recent effort in using EM for imaging chromatin structure is EM tomography with even higher resolution ($1 \times 1 \times 1$ nm)[24], in which photo-oxidized label is used to mark all DNA and to visualize the overall chromatin structures for 3D imaging in the nucleus[25], but within a limited depth at $z$-axis (250 nm). Nonetheless, our ability to visualize 3D chromatin-folding structures in high-quality ultra-resolution remains inadequate, hampered in particular by the lack of sequence specificity, and low depth of imaging in the $z$ dimension.

In our efforts to achieve ultra-resolution visualization of sequence-specific 3D chromatin folding structures, we present 3D electron microscopic in situ hybridization (3D-EMISH) method that combines advanced in situ hybridization using biotinylated DNA probes[26] with silver staining and serial block-face scanning electron microscopy (SBF-SEM)[20,21]. The serial $z$-stack EM images is assembled computationally into a 3D conformation of the targeted genomic region with a resolution of $5 \times 5$ nm in the $xy$ plane and 30 nm in the $z$-axis. We apply 3D-EMISH to visualize a specific chromatin folding location in the human genome, and analyze more than 200 distinctive 3D chromatin structures derived from individual nuclei of human lymphoblastoid cells. Thus, we demonstrate that SBF-SEM can be applied for imaging specific chromatin-folding structures.

## Results

**The 3D-EMISH methodology**. Our method includes 3D preservation of the nuclei, in situ hybridization of specific DNA probes, serial scanning by EM, and imaging data analysis to reconstruct the spatial models of chromatin folding structures (Fig. 1).

In the first step of the 3D-EMISH protocol, cells were fixed with 4% freshly made paraformaldehyde, and then embedded in a thrombin–fibrinogen clot, postfixed, cryoprotected in 30% sucrose, and frozen in dry ice-cooled isopentane. The embedded clots (<5 mm in size) were cut into 40-μm-thick sections in the freezing microtome. The next step in 3D-EMISH is in situ hybridization (ISH). The free-floating 40-μm-thick sections were incubated with the biotinylated DNA probes targeting a specific genomic region of interest. The signals of the biotinylated DNA probes for EM detection were processed with the use of 1.4-nm-thick streptavidin-conjugated fluoronanogold particles and subsequent silver enhancement. The procedure of silver enhancement was experimentally optimized to obtain electron dense 4–5-nm diameter granules as the smallest items (Supplementary Fig. 1a–c).

To optimize chromatin ultrastructure preservation, we performed a set of in situ hybridization (ISH) tests. Using transmission electron microscopy (TEM), we assessed in details the influence of potentially harmful factors associated within the ISH procedure (Supplementary Fig. 1). Comparing with the control of no ISH procedure (nucleus in native state) (Supplementary Fig. 1d), the ISH procedure including permeabilization by Triton X-100 (Supplementary Fig. 1e) or treatment with formamide and high temperature (Supplementary Fig. 1f) showed at most moderate changes or deteriorations in chromatin structure. However, surprisingly, we found that the inclusion of dextran sulfate, commonly used in ISH to increase the probe concentration and the hybridization reaction speed, caused the most distortion to the chromatin ultrastructure (Supplementary Fig. 1g), matched with prior experimental observations[19]. Therefore, dextran sulphate was omitted from ISH in our 3D-EMISH protocol. We observed that the lack of dextran sulphate in ISH had only a minor impact on hybridization efficiency and slightly increased background (Supplementary Fig. 1h–i). To ascertain our observation on dextran sulphate, we performed a specific DNA staining with terminal deoxynucleotidyl transferase (TdT) according to M. Thiry[27]. This method utilizes TdT to add labeled nucleotides to the free DNA ends, formed by ultrathin cutting, that are subsequently detected by immunogold staining. We again observed here that permeabilization with Triton X-100 had caused at most moderate changes or deteriorations in chromatin structure and DNA distribution. The TdT experiments showed further that extensive filamentous objects were observed with the addition of dextran sulfate. Some particles were also found in the cytoplasm, what might suggest the loss of nuclear membrane integrity (Supplementary Fig. 2).

After in situ hybridization, the 40-μm-thick sections were stained with uranyl acetate and tannic acid, followed by dehydrating and flat-embedding in an epoxy resin. Because uranyl acetate stains nucleic acids and proteins broadly, its inclusion in 3D-EMISH enabled us to visualize the overall nuclear space in

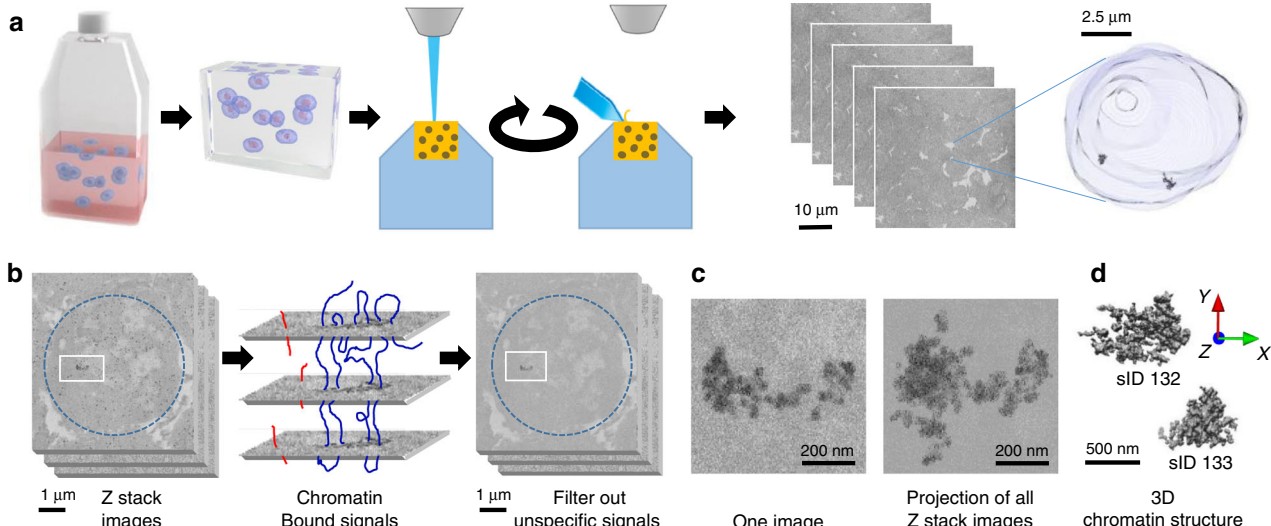

**Fig. 1 3D-EMISH method to visualize ultra-resolution 3D chromatin folding. a** 3D-EMISH schematic. Cells are grown in suspension. After fixation with 4% PFA, thrombin–fibrinogen clot is formed. The clot is postfixed, soaked with cryoprotectant, frozen and cut in 40-μm sections. Free-floating sections are subjected to in situ hybridization with biotinylated DNA probe and processed to SBF-SEM. Then, multiple rounds of ultrathin slicing and imaging are performed. Each cubical sample volume contains dozens of cells. Cell nucleus is segmented, containing two separated target chromatins, as an example in 3D-EMISH. **b** Image processing for 3D-EMISH. First, we searched for the connected components through z-stack images per each identified nucleus (blue dotted circle). Second, the chromatin target structures were identified by removing nonspecific background EM signals. EM signals, connected in multiple consecutive layers were considered as actual target chromatin bound signals, otherwise regarded as chromatin unbound signals, or nonspecific signals. Two scale bars are 1 μm. **c** 3D-EMISH image example of one slice and all z-stack projected image after filtering out background EM signals. Two scale bars are 200 nm. **d** 3D reconstructed chromatin-folding structure examples. We assigned unique structure index number (sID) for each structure; scale bar, 500 nm.

relation to the target chromatin structure (Fig. 1a). One block of the embedded specimens (usually less than 1 mm in width) was placed in the chamber of SBF-SEM (details in "Methods", Zeiss Sigma VP) equipped with a built-in ultramicrotome (Gatan 3View). The specimen block was consecutively sliced one layer at a time at 30–50-nm intervals, and the exposed surfaces of the specimen was serially scanned in a field of $8192 \times 8192$ pixels ($\sim 1700\ \mu m^2$, when a pixel size is 5 nm) to obtain the volumetric data, including the specific 3D-EMISH signals. The specimen was sliced again for the next round of signal acquisition and so forth. This cycling process of slicing–scanning was performed hundreds of times to generate z-stack images in a 3D-EMISH experiment (Fig. 1a).

Since all in situ hybridization reactions result in nonspecific background[26], the 3D-EMISH signals in each slice also included punctate patterns of staining that we deemed likely to be noise (see Fig. 1b; Supplementary Fig. 3). We reason that the specific signals from the targeted chromatin would be in continuation in multiple layers of the z-stack images, whereas the nonspecific signals would not be connected in the z-stacks (Fig. 1b). Therefore, we developed an image-processing algorithm based on multilayer connectivity to retain true chromatin-positive signals and remove false-positive one. The volume ratio of the specific vs. nonspecific signal, in the analyzed regions of interest, was estimated as 0.78 ($\pm 0.40$ STD, $\pm 0.12$ SEM), the nonspecific signal occupied 1.88% ($\pm 0.46\%$ STD, $\pm 0.15\%$ SEM) of the total background volume ($\pm$ STD, 10 different ROIs). Examples of the images before and after background filtering are presented in the Supplementary Fig. 3. After filtering out the background noise, the chromatin signals were assembled to reconstruct the 3D chromatin-folding structure in ultra-resolution for the targeted genomic region (Fig. 1c, d).

**Chromatin-folding structures captured by 3D-EMISH.** We applied 3D-EMISH to investigate chromatin-folding structures

in the genome of the human lymphoblastoid cell line GM12878. This cell line has been extensively studied for epigenome and 3D genome mapping[7,10]. At a specific genomic region (chr7: 141,547,153– 143,237,066, 1.7 Mb length, hg38), three distinctive loop domains were inferred by extensive CTCF ChIA-PET data (Fig. 2a). RNAPII ChIA-PET and RNA-Seq further suggested that the genes in the first loop domain and part of the third domain are in active transcriptional loops. To confirm the presence of the chromatin loop domains inferred by these sequencing-based approaches, we designed and produced biotinylated DNA probes derived from 11 BAC clones to cover this region for 3D-EMISH experiments (Fig. 2a). To test the hybridization efficiency of the probes to the target genomic region, we also generated two-color fluorescence probes from the BAC clones that could help distinguish the three domains (Fig. 2a). Subsequently, we performed 3D FISH imaging analysis using confocal microscopy, and validated that these probes were efficient and highly specific (Fig. 2b).

Using the biotinylated DNA probes, we performed 3D-EMISH experiments in GM12878 cells at the exponential growth phase. We performed two independent experiments using two different EM settings, producing image voxel size of $7 \times 7 \times 50$ nm in replicate 1 and $5 \times 5 \times 30$ nm in replicate 2. The sample cubical blocks were analyzed by a series of 300 (replicate 1) or 600 (replicate 2) slicing and scanning cycles, rendering consecutive 50 nm (replicate 1) or 30 nm (replicate 2) sections, respectively. One specimen, typically about $1000 \times 1000 \times 40\ \mu m$, included tens of thousands of cells. However, the cubic volumes that were scanned by EM microscopy were much smaller and they were 3200 $\mu m^2$ in xy and 15 $\mu m$ in z for replicate 1, and 1700 $\mu m^2$ in xy and 18 $\mu m$ in z for replicate 2. Each of the z-stack images could capture a cross-section of $\sim 10$–30 cell nuclei, intact or truncated (Fig. 2c). After signal de-noising and assembling, the outline of a nuclear framework and the specific chromatin objects could be visualized respectively based on the uranyl acetate and the silver staining signals (Fig. 2d). From two independent 3D-EMISH

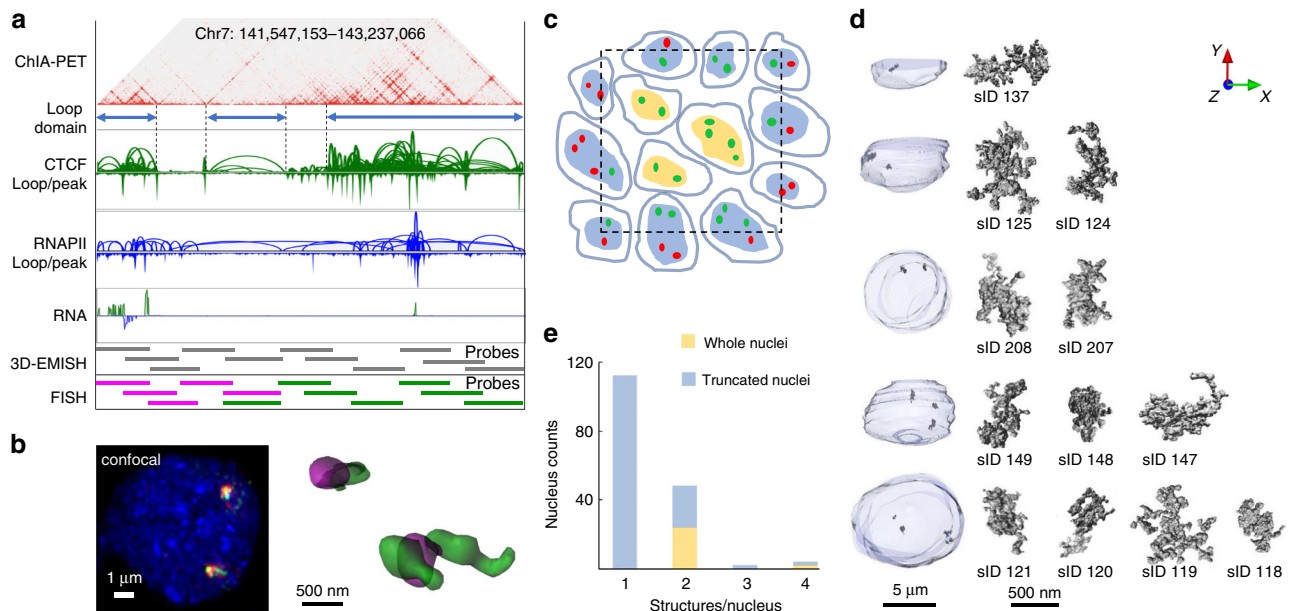

**Fig. 2 3D-EMISH chromatin image collection for 1.7 Mb target genomic region. a** Genome browser view of the target region in hg38, Chr7:141,547,153–143,237,066, in GM12878 cells. From top to bottom, 2D contact map (CTCF ChIA-PET, 5Kb resolution), three distinctive loop domains (141,547,153–141,786,415; 141,975,791–142,304,141; 142,463,980–143,237,066), loop and peak for CTCF ChIA-PET, loop and peak for RNAPII ChIA-PET, RNA-seq, and DNA probes using BAC clone for 3D-EMISH and two-color FISH. **b** Confocal microscopy image of the target genomic region using two-color DNA-FISH (left) and 3D reconstructed FISH signals (right). This experiment was repeated independently three times with similar results. **c** Cross-section illustration of 3D-EMISH cubical samples, where the dotted black square represents the sample boundary. Cells contain either two target chromatins in a nucleus (G1 phase) or four target chromatins in a nucleus (S-M phase). 3D-EMISH collected whole-cell nuclei (W, orange), located at the inner cubic sample, and truncated nuclei (T, blue), located at the outer cubic sample. Intact target chromatin-folding structures (green dots) were collected. Truncated partial structures and structures beyond the cubical sample were not collected, represented as red dots. Total 229 target chromatin-folding structures were collected from two replicates, containing 26 whole nuclei and 140 truncated nuclei. **d** Examples of collected target chromatin-folding structures with their nuclei are presented. From top to bottom, 1(T), 2(T), 2(W), 3(T), and 4(W) structures per a nucleus are shown with structure index numbers (sID, details in Source Data 1 and Supplementary Fig. 5). **e** Nucleus count for 1, 2, 3, and 4 structures per a nucleus.Source Data 1.

experiments, a total of 166 nucleus image stacks were obtained. Most of them (140) were truncated nuclei and contained from 1 to 4 specific target signals, whereas 26 nuclei were intact with two or four specific chromatin targets (Fig. 2e), suggesting that the cells were possibly in the G1 phase (two copies of target region) or S-G2 phase of the cell cycle.

To compare 3D-EMISH with super-resolution microscopy, we applied iPALM[28] and visualized the same targeted chromatin structures using iPALM-specific probes tagged with two-color blinking fluorophores for the same genomic region in GM12878 cells (Supplementary Fig. 4). The iPALM 3D image structure showed about fourfold lower resolution ($20 \times 20$ nm in $xy$) than the 3D-EMISH images. In addition, the depth of an iPALM image at $z$ dimension was limited at 750 nm[29], thus capturing only incomplete chromatin structures, even though isolated nuclei were used for iPALM imaging to reduce the depth of cell specimen. Thus, 3D-EMISH not only provided higher resolution of chromatin images at $xy$ plane than iPALM but also provided greater depth at the $z$-axis than iPALM, enabling visualization of complete 3D chromatin-folding structures. However, the use of two different fluorophores in iPALM allowed determination of the orientation of the chromatin-folding structure along the linear DNA strand.

**Heterogeneity of chromatin-folding structures**. In total, we captured 229 distinct chromatin-folding structures (Supplementary Fig. 5) for the 1.7 Mb region from 166 nuclei examined in our 3D-EMISH experiments. At first glance, it appeared that the chromatin folding structures at this region in GM12878 cells varied from having a single domain to multiple sub-domains. To

further quantitatively analyze the structural features, we developed a computational algorithm to characterize potential chromatin-folding properties. First, we aligned each of the reconstructed 3D chromatin structures along with its principal axes, by orthogonalization of the structure inertia tensor. This representation allows clear dissection of the chromatin signals in more detailed folding structures (Fig. 3a). Next, we calculated the local density centers in each chromatin structure, followed by diffusing the density from the centers to demarcate the boundaries and to identify distinctive domains in the 3D-EMISH structures (Fig. 3a). For example, in the three different 3D-EMISH structures (sID50, sID12, and sID42), one, two, and three local density centers were identified, respectively, corresponding to morphological domains (Fig. 3b; Supplementary Movies 1–3 and Supplementary Fig. 6). The sub-domains in each 3D-EMISH structure were demarcated by different colors (Fig. 3c).

Applying this algorithm to the aforementioned 229 chromatin-folding structures captured by 3D-EMISH, we analyzed the detailed chromatin domain composition in each of the structures (Fig. 4a; Supplementary Fig. 7). Interestingly, we identified 58 (25%) 3D-EMISH structures with one domain, 90 (39%) with two, 70 (31%) with three, and 11 (5%) with four or five domains (Fig. 4b). Only two structures were identified as five domains, and we merged them with four domains structure group for our statistical analysis. Remarkably, the structures with multiple sub-domains accounted for a combined 75% of all the structures, which is approximately in line with the ChIA-PET mapping data (Fig. 2a). To further characterize these chromatin-folding structures, we measured the volume and the surface area for each of them. These measurements showed that the chromatin structures with one domain had the

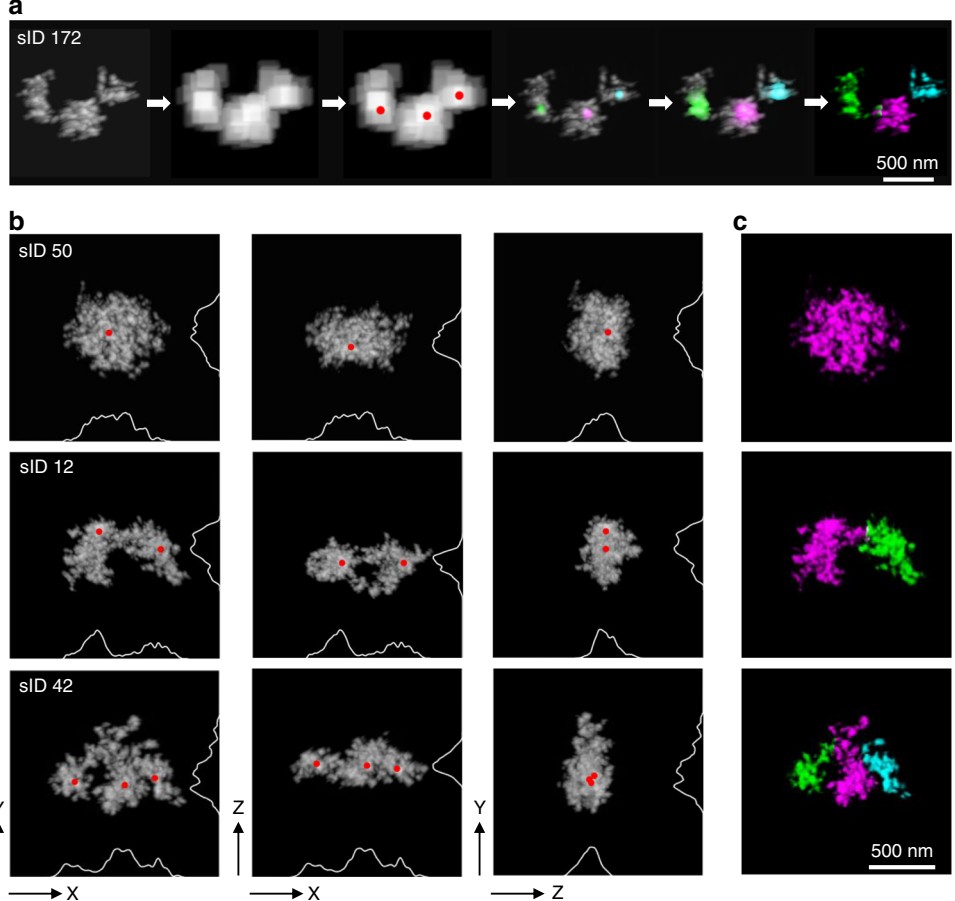

**Fig. 3 3D-EMISH distinctive chromatin-folding domain identification. a** 3D-EMISH image-processing steps to identify distinctive chromatin-folding domain. We smoothed out 3D-EMISH image to remove fine-grain feature, searched for distinctive folded sub-domains, and calculated the center point of each domain, represented as red dots. Applying diffusion process, starting from each center point to cover all voxels in 3D image, we identified each chromatin-folding domain in the 3D-EMISH chromatin image. For example, chromatin structure (sID 172), three local density centers were identified, and then three distinctive chromatin folding domains were identified, presented in three different colors (magenta, green, and cyan); scale bar, 500 nm. **b** Three typical examples of 3D-EMISH chromatin image are shown, single condensed chromatin folding domain (sID50, Supplementary Movie 1), two distinctive chromatin-folding domains (sID12, Supplementary Movie 2), and three distinctive chromatin-folding domains (sID42, Supplementary Movie 3) for three different view angles (xy, xz, and zy from left to right) with identified local density centers (red dots) with projected density signal curves on each axis. **c** Identified individual chromatin-folding domains are distinguished by different colors (magenta for the first domain, green for the second domain, and cyan for the third domain); scale bar, 500 nm.

smallest volume and surface values, and the structures with more than one domain showed increased values along with the numbers of domains (Fig. 4c, d; Supplementary Fig. 8 and Source Data). We also calculated the form factor (based on volume and surface; see Eq. 1 in Methods for the definition) for each structure, which showed the same trend as the other measurements, i.e., single-domain structures had the lowest form factor value (Fig. 4e).

Taken together, our imaging analyses suggest that the small percentage of one-domain structures captured in our data most likely exemplify condensed chromatin-folding states, whereas the majority of the structures contained sub-domains and could represent loose chromatin-folding characteristics with various substructures for different domain functions during interphase in B lymphoblastoid cells. It is noteworthy that many of the structures captured in this 3D-EMISH study showed three domains, consistent with our ChIA-PET data based on bulk cells (Fig. 2a). Furthermore, the ultra-resolution details unraveled a high level of heterogeneity of chromatin-folding structures within this 1.7 Mb region. Strikingly, each individual chromatin ultra-structure, even those having the same number of sub-domains,

showed a uniquely distinctive chromatin-folding conformation (Fig. 4a; Supplementary Fig. 7).

## Discussion

We have demonstrated 3D-EMISH as an effective imaging technique with ultra-resolution ($5 \times 5 \times 30$ nm) capability to capture specific 3D chromatin-folding structures in individual nuclei, superior to the current super-resolution 3D visualization by fluorescence light microscopy. The use of in situ hybridized DNA probes targeting the specific genomic regions provided specificity of folding structures. The use of SBF-SEM enabled single-digit nanometer resolution for the xy dimension. More advantageously, it allowed collecting z-stack images up to 18 μm in total depth, which is not possible by super-resolution microscopy, such as iPALM and STORM, because of their optical image depth limit. In addition, each 3D-EMISH experiment could analyze large numbers of chromatin structures from multiple cells simultaneously, and multiple experiments could be robustly performed for analyzing large numbers of individual cells. The inclusion of en-bloc staining with uranyl acetate used in

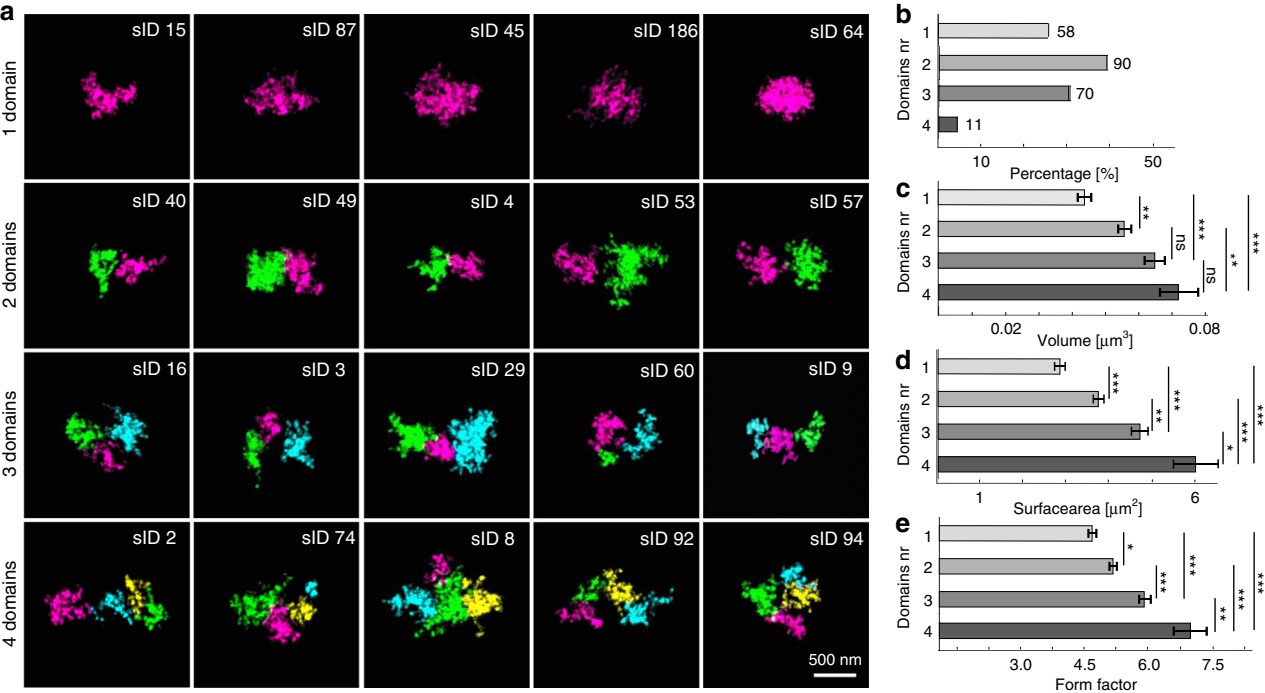

**Fig. 4 3D-EMISH statistical analysis of chromatin-folding domains. a** Classified 1, 2, 3, 4 distinctive chromatin-folding domains examples. Identified chromatin-folding domains in each 3D image were presented by different colors (magenta, green, cyan, and yellow for first, second, third, and fourth domain group, respectively); scale bar, 500 nm. **b** 3D-EMISH chromatin-folding domain histogram of 229 structures: one domain (58/229, 25%), two domains (90/229, 39%), three domains (70/229, 31%), four domains (11/229, 5%). Volume (**c**), surface area (**d**), and form factor (**e**) statistics for all 229 images for four distinctive classified chromatin-folding domains. This experiment was repeated independently two times with similar results. Error bars indicate SEM. Kruskal–Wallis test, and two-sided Mann–Whitney tests with Bonferroni correction were performed; ns (statistically insignificant); statistically significant differences: $*p \leq 0.05$; $**p \leq 0.01$; $***p \leq 0.001$. $P$-values [volume] (1~2: 0.007; 1~3: 0.000; 1~4: 0.000; 2~3: 0.166; 2~4: 0.009; 3~4: 0.261); [surface area] (1~2: 0.000; 1~3: 0.000; 1~4: 0.000; 2~3: 0.002; 2~4: 0.000; 3~4: 0.016); [form factor] (1~2: 0.025; 1~3: 0.000; 1~4: 0.000; 2~3: 0.000; 2~4: 0.000; 3~4: 0.007). Source data are provided as a Source Data file. Data distributions are shown in Supplementary Fig. 8.

3D-EMISH resulted in relatively weak counterstaining of the nuclear area, thus providing additional benefit for outlining the spatial location of the target structure in relation to the nuclear border.

Recently, another EM approach, ChromEMT, has been introduced for chromatin structure analysis[25]. It uses photo-oxidized diaminobenzidine polymers and $OsO_4$ fixation, rendering a detection system size of ~1–2 nm. The method is also characterized by the highest available resolution (<1 nm in *xy*) and, due to utilizing EM tomography, ~1 nm in *z*. Therefore, ChromEMT can visualize single-DNA polymer loops. However, the ChromEMT approach stains all DNA in nuclei, and thus lacks sequence specificity and allows investigation of only 250-nm-thick section. In contrast, 3D-EMISH uses DNA probes to target a given genomic region of interest with high specificity, and the obtained 3D images can be immediately correlated with the available genomic mapping data, thus providing insights into potential structure–function relationships. Nonetheless, these two EM approaches are complementary to each other for ultra-resolution visualization of the global nature of chromatin organization and specific features of chromatin folding.

The ultrastructures of 3D chromatin-folding domains unraveled by 3D-EMISH are ultra-resolution snapshots of the target genomic region from individual cells. All of these 229 snapshots showed a dynamic nature of chromatin folding. The ultra-resolution nature of the 3D-EMISH structures displayed a high level of heterogeneity in chromatin folding with respect to different volume and shape even of the two copies of chromatin fold

in the same nucleus. The variations in the 3D chromatin-folding structures within the 229 images may be a reflection of spatio-temporal dynamics and potential functional properties such as transcription activity and epigenetic state[15], and also cell cycle phase[25,30]. Heterogeneity in the chromatin organization is observed at different levels. For example, we have shown different conformations of chromosome 1 even within the same nucleus[10]. There are also results obtained in human cells showing high variability in the physical distances between selected genomic loci[31] and variability in the organization of chromatin domains[16]. Moreover, studies of transcription reveal expression heterogeneity[32]. Also in the same cells that we studied—GM12878, heterogeneity was shown using single cell sequencing approaches —scATAC-Seq[33] and scRNA-Seq[34] when compared with bulk studies. It suggests that variability observed by us and others could interplay with transcriptional activity. Nevertheless, the relation between dynamic changes of chromatin structure and stochasticity in gene expression is not yet fully understood[35,36].

## Methods

**Cell culture.** GM12878 (human lymphoblastoid) cells were purchased from commercial cell line, Coriell Institute. These cells were cultured in RPMI-1640 medium (Sigma) with addition of 2 mM GlutaMax (Gibco), 15% FBS (Gibco), 100 U/ml penicillin and 100 μg/ml streptomycin (Gibco), at 37 °C; 5% $CO_2$. The cells authentication was done by the Jackson Laboratory for the Genomic Medicine according to ATCC (American Type Culture Collection) recommended authentication tests: Morphology Check by optical observation of a magnified cell culture and Growth Curve Analysis. Any cells showing inconsistent growth properties were discarded.

**DNA probes used in in situ hybridizations**. The DNA probe was designed based on BAC (bacterial artificial chromosome) clones obtained from CHORI (Children's Hospital Oakland Research Institute). The designed 11 BACs (CH17-148N23, CH17-120P14, CH17-417P16, CH17-265E23, CH17-52M16, CH17-310K12, CH17-121M24, CH17-326A22, CH17-215B18, CH17-450A09, and CH17-227I19) covered our genomic target region in human chromosome 7: 141,547,153–143,237,066, in hg38. The clones were verified with PCR using primers listed in Supplementary Table 1.

After isolating DNA from BACs using alkaline lysis method, DNA amplification was performed using the Illustra *GenomiPhi* V2 DNA Amplification Kit (GE Healthcare Life Sciences) and DNA labeling with biotin using the Biotin-Nick Translation Mix (Roche) and digoxigenin with the DIG-Nick Translation Mix (Roche). The product of reaction was mixed with human competitor DNA (Applied Genetics Laboratories, Inc.), salmon sperm DNA (Invitrogen), absolute ethanol and incubated 2 h at −20 °C. Then, the probe was centrifuged, dried with a vacuum centrifuge, and dissolved in 100% formamide (overnight; 37 °C; with shaking). To obtain a ready-to-use probe, a 4× SSC buffer was added to the probe dissolved in 100% formamide (1:1; finally probe dissolved in 50% formamide in 2× SSC). In the case of light microscopy, dextran sulphate was also added to final concentration 10%.

**Cell preparation for confocal microscopy**. The cell pellet was fixed with 4% paraformaldehyde in PBS (10 min; RT), washed with PBS and cells were spun down onto a glass slide ($1 \times 10^5$ cells/13-mm coverslip) using a cytocentrifuge.

**Cell preparation for iPALM**. Nuclei were isolated (Sigma; Nuclei Isolation Kit: Nuclei EZ Prep), fixed with 4% paraformaldehyde in PBS (10 min; RT) and attached onto coverslips (with fiducial markers; see: iPALM imaging and post processing in "Methods") washed earlier with 1 M KOH (20 min), rinsed with water and coated with 0.01% poly-L-Lysine for 20 min (Sigma).

**DNA-FISH protocol**. 3D DNA-FISH was performed according to Cremer et al.[37], with modifications. In details, after blocking of endogenous biotin (Vector Laboratories; SP-2001), the cells were permeabilized with 0.5% Triton X-100 (Sigma) in PBS (RT, 20 min). To augment permeabilization, the cells were immersed in a cryoprotectant solution (20% glycerol in PBS, RT, 2 h) followed by their repeated freezing–thawing above the surface of the liquid nitrogen ($4 \times 30$ s). Subsequently, the cells were washed in PBS containing 0.05% Triton X-100 (Sigma) (RT, 3 times for 5 min), treated with 0.1 N HCl (RT, 5 min), washed again in 2× SSC at 37 °C and incubated in 50% formamide in 2× SSC (4 °C, overnight). On the next day, the probe was added, the samples were denatured (5 min, 80 °C) and hybridized (48 h, 37 °C). Then, the samples were washed with 2× SSC ($3 \times 5$ min; 37 °C; with shaking), 0.1× SSC ($3 \times 5$ min; 60 °C; with shaking), rinsed briefly with 0.2% Tween (Sigma) in 4× SSC and incubated with blocking solution—4% BSA (Sigma), 0.2% Tween (Sigma), 4× SSC (1 h, 37 °C) following incubation with the detection system for confocal microscopy or iPALM.

**DNA-FISH signal detection for confocal microscopy**. DNA probe signals were captured by two-steps system detection of biotin-labeled probe with avidin–AlexaFluor488 (Molecular Probes; dilution 1:100; final concentration: 10 µg/ml) and anti-avidin-FITC (Sigma; dilution 1:100; final concentration: 20 µg/ml), and two-steps system detection of digoxigenin-labeled probe with anti-digoxigenin–rhodamine produced in sheep (Roche; dilution 1:100; final concentration: 2 µg/ml) and secondary antibody donkey anti sheep-TRITC (Jackson ImmunoResearch; dilution 1:100; final concentration: 10 µg/ml). DNA was counterstained using Hoechst 33342 (Molecular Probes), and samples were mounted with Vectashield (Vector Laboratories).

**Confocal microscopy imaging and post processing**. The samples were imaged with Zeiss LSM 780 microscope, using a 63× oil immersion objective (NA 1.4) and a 405-nm diode laser, 488 nm line of argon laser, 561-nm DPSS diode laser, 633-nm HeNe laser; with 70-nm *xy* pixel size, and 210-nm *z*-spacing. The image stacks were deconvolved using Huygens software (SVI) with the maximum-likelihood estimation (MLE) algorithm. FISH signals imaged by confocal microscopy were segmented based on the threshold and reconstructed using Imaris (Bitplane) software.

**DNA-FISH signal detection for iPALM**. DNA probe signals were captured by one-step system detection of biotin-labeled probe with streptavidin-CAGE590 (Abberior; dilution 1:250; final concentration: 3.5 µg/ml), and one-step system detection of digoxigenin-labeled probe with anti-digoxigenin–Alexa Fluor 647 (Jackson ImmunoResearch; dilution 1:250; final concentration: 2 µg/ml).

**iPALM imaging and post processing**. Coverslips for iPALM with fiducial markers[29] were prepared according to the protocol: 25 mm diameter #1.5 thickness high-precision coverslips (Warner Instruments) were cleaned in a basic "piranha" solution of 5:1:1 $H_2O$:$NH_3OH$:$H_2O_2$ (50% w/v) at 90 °C for 4 h, followed by washing in deionized water and drying. Coverslips were then coated with 5 mg/ml

poly-L-Lysine (PLL), MW 70,000 (MP Biomedicals) in water for 30 min, and washed. Finally, a solution of negatively charged gold nanorods with major/minor diameters of 60/40 nm (Nanopartz, Inc.) was allowed to adhere to the PLL layer, followed by further washing of excess particles and drying. Finally, coverslips were coated with a ca. 50-nm layer of $SiO_2$ to passivate the fiducial markers using a Denton Vacuum Explorer sputter coater. The samples for iPALM were imaged in STORM buffer[38], containing 50 mM TRIS (pH 8.0), 10 mM NaCl, 0.5 mg/ml glucose oxidase, 40 µg/ml catalase, 10% glucose (w/v), and 100 mM 2-mercaptoethylamine (MEA) (all from Sigma), using coverslips with fiducial markers[29] described earlier in this paragraph. Above cells deposited on the coverslip with fiducial markers, second clean coverslip was placed, sealed and samples were imaged using noncommercial iPALM system[28] available at Advanced Imaging Center–HHMI's Janelia Research Campus, with excitation: 647 nm laser (5.5 kW/cm$^2$) and 561-nm laser (4.5 kW/cm$^2$) and activation: 405-nm laser (100 W/cm$^2$). The number of frames was 25,000, and the exposure time was about 50 ms. iPALM data were processed to extract single-molecule 3D coordinates and localization precisions, and subsequently visualized using the PeakSelector software (Janelia Research Campus)[29]. FISH signals acquired by iPALM were segmented based on the threshold with PartSeg software (https://4dnucleome.cent.uw.edu.pl/PartSeg/). Then, 3D reconstructions of iPALM images were obtained using Imaris (Bitplane).

**Cell preparation for 3D-EMISH**. The experiments were performed on cultured GM12878 cells. After centrifugation, fixation of the cell pellet ($\sim 7 \times 10^7$ cells) with 4% paraformaldehyde in PBS overnight at 4 °C and washing with PBS, the artificial tissue with embedded cells was formed. To the fixed and washed pellet, 0.2 ml of the fibrinogen solution (100 mg fibrinogen (Sigma; F3879), 53.33 mg of sodium citrate, 283.33 mg of sodium chloride, 33.33 ml $H_2O$) was added and stirred. After centrifugation, 0.2 ml the thrombin solution (Sigma; T7009-100UN + 1 ml $H_2O$) was added[39]. The clot was postfixed with 4% paraformaldehyde in PBS (10 min; RT) and washed with PBS. Then, the formulated artificial tissue was soaked with 30% sacharose in PBS overnight at 4 °C and frozen in tissue freezing medium (Leica). After overnight storage at −80 °C, the clot was cut into 40-µm slices using cryostat (Leica).

**In situ hybridization for 3D-EMISH**. In situ hybridization was performed on the free-floating sections of the clot according to Cremer et al.[37], with modifications. In details, the sections were washed with PBS ($3 \times 5$ min; RT), 0.5% Triton X-100 (Sigma) in PBS ($1 \times 20$ min; RT), 2× SSC ($2 \times 2$ min; RT), incubated with 0.1 mg/ml RNase A in 2× SSC (Sigma; R6513) (10 min; 37 °C), washed with 2× SSC ($2 \times 2$ min; 37 °C), incubated with citrate buffer pH 6 (30 min; 80 °C), cooled down to room temperature and washed with 2× SSC ($2 \times 5$ min; RT). After that, blocking of endogenous biotin was performed using the Avidin/Biotin Blocking Kit (Vector Laboratories; SP-2001), by incubation with Avidin D (four drops of the Avidin D solution to 1 ml 2× SSC) (15 min; RT), briefly washing with 2× SSC and incubation with the biotin solution (four drops of the biotin solution to 1 ml 2× SSC) (15 min; RT), followed by washing with 2× SSC ($2 \times 5$ min; RT) and incubated with 50% formamide in 2× SSC overnight at 4 °C. After this step, the sections and biotin-labeled DNA probes were overlaid with a glass chamber, sealed with rubber cement, prehybridized (4 h, 45 °C), denatured (5 min, 80 °C) and hybridized (48 h, 37 °C). After 2-days hybridization, sections were washed with 2× SSC ($2 \times 10$ min; 37 °C; with shaking), 0.1× SSC ($2 \times 10$ min; 60 °C; with shaking), incubated with blocking solution- 4% BSA (Sigma), 0.2% Tween (Sigma), 4× SSC (1 h, 37 °C) following incubation with Alexa Fluor$^{TM}$ 488 FluoroNanogold$^{TM}$-Streptavidin (Nanoprobes; 7216) in 2% BSA (Sigma), 0.2% Tween (Sigma), 4× SSC (overnight; 4 °C; with shaking—concentration 0.8 µg/ml) and washed with 0.2% Tween (Sigma) in 4× SSC ($3 \times 10$ min, RT; with shaking). After hybridization and washing, sections were mounted in 2× SSC buffer and imaged with the Zeiss LSM 780 confocal microscope using the 488-nm argon laser and the water objective (C-Apochromat 40×/1.2 W Corr. FCS). Only sections containing nuclei with well-visible FISH signals were processed to 3D-EMISH.

**3D-EMISH staining**. Sections after passing the quality control using confocal microscopy were washed with $ddH_2O$ ($5 \times 3$ min; RT) and incubate with silver enhancement kit to increase size of 1.4 nm nanogold particles ($4 \times 10$ min; RT; LI Silver Enhancement Kit (Molecular Probes; L-24919)) and washed tightly with $ddH_2O$. After silver enhancement, sections were postfixed with 2.5% glutaraldehyde in 0.1 M cacodylate buffer (Sigma) (2 h; RT), washed with 0.1 M cacodylate buffer (Sigma) ($3 \times 10$ min; RT), incubated with 1% tannic acid in 0.1 M cacodylate buffer (Sigma) ($2 \times 2$ h; 4 °C), washed with water ($5 \times 3$ min; RT) and incubated with 4% uranyl acetate in 50% ethanol (overnight; RT). Then the dehydration was performed using increasing concentration of ethanol: 50%, 60%, 70%, 80%, and 95% ($1 \times 15$ min each; RT), absolute ethanol ($2 \times 20$ min; RT). After dehydration, sections were prepared to embedding by incubation with acetone ($2 \times 10$ min; RT), increasing the ratio of Epon (Sigma)/acetone mix: 1:3, 1:1, 3:1 (1 h each; RT), and Epon (1 h; RT and then overnight; RT). The next day, Epon was replaced with fresh one, and specimens were embedded between two sheets of ACLAR® Film and placed into 60 °C oven (24–48 h). After polymerization of the resin, the sample was cut with razor blade and scalpel, glued on pin (metal rivet; Oxford Instruments) with silver conductive resin (CircuitWorks), and incubated 2 h at 65 °C. Using

ultramicrotome (Leica) and diamond knife (DiATOME), smooth surface of sample to SBF-SEM was obtained. Then, to enhance conductivity, sides of the sample were painted with silver paint (Ted Pella).

**EM imaging**. To image 3D-EMISH samples, Zeiss Sigma electron microscope with backscatter electron detector and ultramicrotome with diamond knife inside the chamber was used (3View2 from Gatan). All images were collected using variable pressure mode with EHT 4–6 kV, aperture 30 μm and resolution $8192 \times 8192$ pixels. Replicate 1 was collected with pixel size 7 nm, sliced each 50 nm, 300 times (voxel size: $7 \times 7 \times 50$ nm). Replicate 2 was collected with pixel size 5 nm, sliced each 30 nm, 600 times (voxel size: $5 \times 5 \times 30$ nm). To examine potential harmful effect of reagents on nuclear ultrastructure, a transmission electron microscope JEM 1400 (JEOL) was used.

**Image processing and quantitative analysis**. All 3D electron microscopy stacks were preprocessed with ImageJ plugin–Linear Stack Alignment with SIFT[40]. The images were manually inspected, and then a cuboid containing the region of interest (ROI) was cut. The mask defining each ROI was determined by connecting component analysis of objects inside the cuboid. We segmented the largest connected component using custom-made software (3D-EMISH image-processing code), see the "Code Availability"), where we used Maximum Entropy algorithm (ImageJ plugin) to optimize the threshold value to segment the structure. The resulting mask was overlaid on the image to segment the structure. Subsequently, the 3D images were resampled in order to obtain the same voxel size (5 nm) in $x$, $y$, and $z$. Reslicing of the 3D images was performed by upsampling with linear interpolation between adjacent planes. We also applied a gaussian filter of size 1 pixel (i.e., 5 nm) in the $x$–$y$ planes in order to remove possible pixel noise. The isotropic scale was required by the plugin used to produce the movies (with unequal scales the structures appear in the movies unnaturally flattened), and also it was needed by the algorithm for morphological domains separation, which operates on a cubical grid. The upsampling is not obviously visible unless we rotate the structure through an angle perpendicular to the axial direction through an axis perpendicular to its longest (principle) axis, which in general is randomly oriented with respect to the slicing direction (Supplementary Movies). Then, Gaussian filter (5 nm cutoff size) was applied to each $z$-section to remove the detector noise. The volume ($V$) and surface ($S$) of the segmented structure were calculated by 3D Object Counter plugin (ImageJ). The form factor ($ff$) was defined as

$$ff = (36\,\pi)^{-\frac{1}{3}}\,S\,V^{-\frac{2}{3}}, \tag{1}$$

where the value is 1 for a spherical object and increase as the structure is less compact than a sphere. For illustrative purpose, the segmented structures were aligned accordingly to their principal axis[41]. The structures were classified into five different morphological groups: (1) compact sub-domain group (one spherical ball shape), (2) distinctive sub-domain group (dumbbell shape) and (3–5) distinctive sub-domain groups, by applying the following sub-domain identification algorithm as follows:

First, the structure was smoothed out by applying the maximum filter with 135 nm cutoff size. This parameter setting was experimentally tested and determined to prevent from over-segmenting sub-domains of a structure, yet preserving the overall morphological feature. Second, the local maximal density centers were calculated per each identified distinctive group. Third, these local maxima were subsequently used as seed points in computing of the diffusion process, where the image density was used as a local diffusion coefficient. The diffusion process was simulated by solving numerically an uncoupled set of diffusion equations,

$$\partial u_i(\mathbf{r}, t)/\partial t = \nabla \cdot [D(u_i(\mathbf{r}, t), \mathbf{r})\nabla u_i(\mathbf{r}, t)], \tag{2}$$

where $u_i$ denotes the density of the diffusing material, and $D$ denotes the EMISH signal density, $i = 1 \ldots N$ (the number of seed points), $\mathbf{r}$ are the pixel coordinates in 3D space.

Initially, we assume that all EMISH signals of each sub-domain was concentrated in each seed point. We represented it as normalized gaussian functions centered at the seed points. Image voxels, diffused from the same seed point, belong to the same sub-domain group. The 3D image region of low EMISH signal density will diffuse slower than that of high EMISH signal density. The boundary voxels had diffused signals from two or more different seed points, and these boundary voxels belong to the sub-domain group which is the highest diffused signal. The diffusion process was completed when all voxels filled by a nonzero density and the structure was divided into separate sub-domains.

The specific-to-nonspecific signal ratio was measured for a set of ten randomly selected three-dimensional region of interests (ROIs), each of them was an EM stack, containing a single structure and nonspecific background. We provided a mean signal ratio, its standard deviation, and a standard mean error, these values were calculated over a set of selected ROIs. The signal ratio was calculated as follows: For each of these 3D EM stacks, we created the following binary masks: specific signal mask (SP), nonspecific background signal mask (NS), and no-signal background mask (BN). The specific and nonspecific signal volume were obtained by summing all mask elements for SP and NS masks, respectively, and multiplying them by the voxel volume, by which we obtain specific-to-nonspecific signal ratio.

The content of the nonspecific signal in the background volume was obtained by dividing NS mask volume by the total background volume (sum of NS and BN mask volumes)

**Data collection and 3D visualizations**. For the data collection, we used Fiji distribution of ImageJ (https://imagej.net/Fiji/Downloads#Life-Line_Fiji) and Gatan Microscopy Suite Software (commercially available). The images of nuclei were cropped manually, 3D visualizations (Figs. 1, 2; Supplementary Fig. 3) were prepared using Amira (FEI), Imaris (Bitplane), PartSeg (https://4dnucleome.cent.uw.edu.pl/PartSeg/) and UCSF Chimera software[42]. Data were analyzed using custom written scripts, see "Code Availability".

**Statistical analysis**. Statistical analysis was performed using IBM SPSS Statistics software. In all cases, error bars indicate SEM. The morphological feature statistics of 3D-EMISH structures for different sub-domain groups was performed using Kruskal–Wallis test with two-sided Mann–Whitney tests with Bonferroni correction (post hoc tests). Statistically insignificant comparison was marked as ns, and statistically significant differences were marked with asterisks: $*p \leq 0.05$; $**p \leq 0.01$; $***p \leq 0.001$.

**Reporting summary**. Further information on research design is available in the Nature Research Reporting Summary linked to this article.

## Data availability
The 3D-EMISH image and data files are available at the following public repository server: https://github.com/3DEMISH/3D-EMISH. Source data for the figures presented in this manuscript are available in the Source Data file. All other relevant data are available from the authors upon reasonable request.

## Code availability
The 3D-EMISH image-processing code is available at the following public repository server: https://github.com/3DEMISH/3D-EMISH.

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

## Acknowledgements

This work was primarily supported by the Human Frontier Science Program, grant no. RGP0039/2017 (to Y.R. and G.M.W.). Y.R. is supported by NIH UM1 (HG009409, ENCODE), U54 (DK107967, 4DN), HFSP (RGP0039/2017), and the Roux family endowment. G.M.W. is also supported by the grants from Polish National Science Center no. UMO-2013/08/M/NZ3/00655. P.T. is also supported by a Polish National Science Centre Grants 2014/15/N/NZ2/00379 and 2017/24/T/NZ2/00307. Y.R. and B.L. are further supported by 4DN (U54 DK107967), ENCODE (UM1 HG009409) consortia, and the Roux family endowment. D.P., G.B., M.K., and Z.P. are supported by Polish National Science Centre (2014/15/B/ST6/05082), Foundation for Polish Science co-financed by the European Union under the European Regional Development Fund (TEAM), and 4DN (U54 DK107967). K.K.P. is partially supported by ETIUDA grant from Polish National Science Centre no. UMO-2019/32/T/NZ4/00502. A.M is partially supported by Sonata bis grant from Polish National Science Centre UMO-2015/18/E/NZ3/00730. The iPALM imaging work was conducted in collaboration with the Advanced Imaging Center at Janelia Research Campus of Howard Hughes Medical Institute. The authors would like to thank Emil Drzewek for his help with Fig. 1a (SBF-SEM scheme).

## Author contributions

G.M.W. and Y.R. conceived and supervised the project. P.T., B.R., and B.L. designed the experiments. P.T. performed the 3D-EMISH experiments under G.M.W.'s supervision, and generated the raw imaging data with K.K.'s assistance. B.R. developed the sub-domain identification algorithm and processed them. P.T., B.R., K.K.P., G.B., and K.K. interpreted and processed the raw data. B.R., K.K.P., G.B., and M.K. performed post processing of EMISH images, and/or statistical analysis. B.R. and K.K.P. designed and performed the detection of morphological domains in the EMISH-detected structures. J.A. executed iPALM imaging, together with P.T. and G.M.W. M.S. and A.A.S. participated in EMISH imaging. Artur Wolny contributed toward the paper figures preparation S.A. and M.K.A. help in electron microscopic imaging. Agnieszka Walczak participated in the initial design of EMISH hybridization. A.M., Z.P., and D.P. provided remarks to the discussion on experimental strategy. P.T. wrote the first draft of the paper. Y.R., B.L., G.M.W., B.R., P.T., and K.K.P. extensively edited the paper. All authors revised the final paper.

## Competing interests

The authors declare no competing interests.
