## [Peer Review File · Nature Communications]

Reviewers' Comments:

Reviewer #1:

Remarks to the Author:

The work by Wilczynski et al. describes an electron microscopy (EM) approach suitable for in situ analysis of chromatin/genome organization at specific genomic loci. Firstly, the DNA-DNA in situ hybridisation on 40 µm of thawed and permeabilized cryosections of human lymphoblastoid cells is done. The probes are then localized by biotin-streptavidin interaction and ultra-small gold particles followed by silver enhancement. The EM imaging is performed using serial block-face scanning EM where the freshly cut block face of in situ hybridized and resin-embedded cells is imaged by detection of back-scattered electrons. Computer 3-D reconstruction of obtained Z-stack images is applied.

This manuscript is clearly written and in detail. The methodological data are well documented by schemes, micrographs, and statistic evaluation and I do not have not any major reservations. I believe this work will interest many workers in the field of the structure and function analysis of chromatin.

Comments and suggested improvements:

1. The important question concerns the principal novelty of the reviewed paper compared with the papers published by Ou et al. (ref 19) or Rouquette et al. (ref 21) or by Hoang et al. 2017, JSB 197:123-134 (should be included). Similarly, I do not consider 5 x 5 x 30 nm resolution as "ultra-resolution" or "unprecedented resolution" at the level of EM. However, the last sentence of the Discussion correctly points out the complementarity of such works. I absolutely agree – the issue is that this work is not really a novel EM method – it is important but rather complementary to others. "Only" novelty represents the combination of pre-embedding EM in situ hybridisation with serial block-face scanning EM.
2. The next important question concerns the potential deleterious effect of reagents including penetration or denaturation treatment on nuclear ultrastructure as well as on antigen destroying. Unfortunately, the figures from transmission EM are very small and were probably taken with a low-resolution camera. However, it seems that there are some differences between the fine structure (Suppl. Fig. 1d-f). For example, the nucleoplasm on suppl. Fig. 1e is more homogenous compared with Fig. 1d. Moreover, it is not easy to pinpoint the location of chromatin. Also, „the dramatic changes of chromatin ultrastructure" (red arrow heads on Fig. 1g) should be specified. The effect of in situ hybridisation using Triton X-100 and formamide on thawed cryosections should be re-evaluated. Immunolabeling with anti-DNA antibody (similarly to the work of Solovei et al., ref. 18) or specific staining of chromatin should be help with this question.
3. It is a pity that the en-block staining with uranyl acetate only does not enable to visualise the target chromatin structure in relation to large-scale chromatin organization as to see in Fig 1. Again, stronger or chromatin-specific staining could help strengthening the work.
4. The Discussion should be enlarged by the biological aspect of obtained results, at least. There is only one sentence talking about the fact that the variations in the 3D chromatin folding may be a reflection of transcription activity, epigenetic state and cell cycle phase. To see transcription or even replication activity on labelled chromatin structures, it suggests itself to perform an effect experiment with incorporation of labelled uridine and/or deoxyuridine followed by immunolocalization with antibodies coupled with fluorochromes or colloidal gold particles. But I understand that this probably it is beyond the work scope.
5. Concerning non-specific background, how big of a part of the overall signal was represented by non-specific background? Could the authors provide one image without any image filtering?

Reviewer #2:

Remarks to the Author:

The authors describe the use of serial block face scanning EM to visualize chromatin folding structure. They applied their method to image and visualize a specific chromatin folding location in the human genome and furthermore analyzed more than 200 different chromatin folding

structures of human lymphoblastoid cells.

The authors describe a method that can on the one hand deliver the resolution that is needed for visualizing the fine structure of chromatin and which on the other hand is not limited in imaging depth, so that whole chromatin folding structures can be acquired. They place their work therefore between super-resolution fluorescence imaging (STORM, achievable resolution 20 x 20 x 50 nm, limited in imaging depth, sequence-specific) and EM tomography (1 x 1 x 1 nm, but limited imaging depth and sequence-unspecific). The method of the authors is capable of achieving a resolution of 5 x 5 x 30 nm, is practically unlimited in imaging depth and sequence-specific.

The paper is well structured and written in a very clear way, and I appreciate the effort with which figures and supplementary videos were prepared. The introduction nicely describes the state of the art and why the described work is needed and how it advances chromatin visualization.

Here some questions and points which I'd recommend to address:

(1) Mention explicitly that this is the first time serial block face scanning electron microscopy is used for imaging and visualizing chromatin structure.

(2) Mention in the introduction that EM tomography can achieve a resolution of 1 x 1 x 1 nm. In my opinion, this is really important. The authors mention a couple of times that they are imaging chromatin with an "unprecedented resolution" (l. 24 and 75) and only later in the discussion the reader learns that previous work achieved a much higher resolution. So I'd recommend to omit the "unprecedented resolution", mention the 1 x 1 x 1 nm resolution of EM tomography in the introduction, so that it is from the beginning on clear that it is not only about resolution, and highlight more the other advantages of 3D-EMISH like its sequence specificity.

(3) The authors show chromatin images acquired with iPALM to compare their method with super-resolution microscopy. It would be great (if feasible) to also see images acquired with EM tomography for comparison.

(4) From the supplementary movies it is not obvious that the axial resolution is 6 times worse than the lateral resolution. Did the authors process the data in any way in this regard?

(5) The provided Github link is not working. I assume that the repository is still private and will be made public upon publication, but still wanted to mention this.

POINT BY POINT RESPONSE TO REFEREES' COMMENTS

Manuscript ID: [NCOMMS-19-33009-T]

We thank the reviewers for their encouraging comments and insightful suggestions. We really appreciate their time and efforts. Please find our point-by-point responses to the reviewers' comments below. For clarity, the reviewers' comments are in *italic blue* while our responses are in black.

Reviewer 1

The work by Wilczynski et al. describes an electron microscopy (EM) approach suitable for in situ analysis of chromatin/genome organization at specific genomic loci. Firstly, the DNA-DNA in situ hybridisation on 40 µm of thawed and permeabilized cryosections of human lymphoblastoid cells is done. The probes are then localized by biotin-streptavidin interaction and ultra-small gold particles followed by silver enhancement. The EM imaging is performed using serial block-face scanning EM where the freshly cut block face of in situ hybridized and resin-embedded cells is imaged by detection of back-scattered electrons. Computer 3-D reconstruction of obtained Z-stack images is applied. This manuscript is clearly written and in detail. The methodological data are well documented by schemes, micrographs, and statistic evaluation and I do not have not any major reservations. I believe this work will interest many workers in the field of the structure and function analysis of chromatin.

We thank the reviewer for the positive note on our manuscript, and appreciate valuable comments.

Comments and suggested improvements:

1. The important question concerns the principal novelty of the reviewed paper compared with the papers published by Ou et al. (ref 19) or Rouquette et al. (ref 21) or by Hoang et al. 2017, JSB 197:123-134 (should be included). Similarly, I do not consider 5 x 5 x 30 nm resolution as “ultra-resolution” or “unprecedented resolution” at the level of EM. However, the last sentence of the Discussion correctly points out the complementarity of such works. I absolutely agree – the issue is that this work is not really a novel EM method – it is important but rather complementary to others. “Only” novelty represents the combination of pre-embedding EM in situ hybridisation with serial block-face scanning EM.

We agree with the reviewer's comment. Indeed, the principal novelty of our approach is substantiated by the combination of SBF-SEM and ISH, which enabled us to visualize the specific target genomic segment with EM resolution that is superior to the current super-resolution of chromatin structures by fluorescent light microscopy such as iPALM or STORM.

We apologize for the lack of clarity in our use of “ultra-resolution” in our manuscript. The meaning of “ultra-resolution” is in comparison to the often phrased “super-resolution” by fluorescent light microscopy for imaging chromatin folding structures. We have reworded this part in the revised manuscript accordingly to avoid mis-understanding.

We also agree with the reviewer that “unprecedented resolution” might be an overstatement and unnecessary. We therefore removed this word in our revised manuscript.

2. The next important question concerns the potential deleterious effect of reagents including penetration or denaturation treatment on nuclear ultrastructure as well as on antigen destroying. Unfortunately, the figures from transmission EM are very small and were probably taken with a low-resolution camera.

We agree with the reviewer's comment. The loss of resolution was a technical glitch that occurred due to inappropriate image formatting, which resulted the low resolution of the images. In the revised manuscript we corrected the problem and maintain the resolution of the original micrographs

However, it seems that there are some differences between the fine structure (Suppl. Fig. 1d-f). For example, the nucleoplasm on suppl. Fig. 1e is more homogenous compared with Fig. 1d. Moreover, it is not easy to pinpoint the location of chromatin. Also, „the dramatic changes of chromatin

ultrastructure” (red arrow heads on Fig. 1g) should be specified. The effect of in situ hybridization using Triton X-100 and formamide on thawed cryosections should be re-evaluated.

We thank the reviewer for this comment. Indeed, in our mock ISH experiments, we observed the negligible deterioration of chromatin structure after Triton X-100 and some changes after formamide plus high temperature. Nevertheless, the most deleterious effect was caused by dextran sulfate, which severely affected the normal chromatin architecture.

In the revised manuscript, we included updated information in the Results (page 5) as: “Using transmission electron microscopy (TEM), we assessed in details the influence of potentially harmful factors associated within the ISH procedure (Supplementary Fig. 1). Comparing to the control of no ISH procedure (nucleus in native state) (Supplementary Fig. 1d), the ISH procedure including permeabilization by Triton X-100 (Supplementary Fig. 1e) or by formamide and high temperature (Supplementary Fig. 1f) showed negligible changes in chromatin structure. However, surprisingly, we found that the inclusion of dextran sulfate, commonly used in ISH to increase the probe concentration and the hybridization reaction speed, caused the most distortion to the chromatin ultrastructure (Supplementary Fig. 1g), similar to what reported by Solovei et al.¹⁹. Therefore, dextran sulphate was omitted from ISH in our 3D-EMISH protocol. We observed that the lack of dextran sulphate in ISH had only a minor impact on hybridization efficiency and slightly increased background (Supplementary Fig. 1h-i).”

Please see the updated Supplementary Fig. 1 and the figure legend in the revised manuscript.

Immunolabeling with anti-DNA antibody (similarly to the work of Solovei et al., ref. 18) or specific staining of chromatin should be help with this question.

According to the Reviewer’s suggestion, we performed a specific DNA staining with terminal deoxynucleotidyl transferase according to M. Thiry’s protocol (The Journal of Histochemistry and Cytochemistry, Highly Sensitive Immunodetection of DNA on Sections with Exogenous Terminal Deoxynucleotidyl Transferase and Non-isotopic Nucleotide Analogues’ indicating the localization of chromatin. The Journal of Histochemistry and Cytochemistry Vol. 40, NO. 3, pp. 411-419. 1992). As shown in the new Supplementary Fig. 2, the experiment confirmed our previous observation, similar to the observation by Solovei et al (ref. 19 in revised manuscript). We found that permeabilization with Triton X-100 causes negligible changes in chromatin structure and DNA distribution. In our hands, formamide/high temperature treatment did evoke some damage to the chromatin. However, addition of dextran sulfate did cause the most severe abnormality in chromatin structure represented by the formation of filamentous meshwork. The meshwork was overlaid by colloidal gold particles indicating the DNA location in the nucleus. Some particles were also found in the cytoplasm. The two latter findings are in agreement with *Solovei et al.* In the revised manuscript, we added in page 5: “To ascertain our observation on dextran sulphate, we performed a specific DNA staining with terminal deoxynucleotidyl transferase (TdT) according to M. Thiry²⁷. This method utilizes TdT to add labelled nucleotides to the free DNA ends, formed by ultrathin cutting, that are subsequently detected by immunogold staining. These TdT experiments showed that extensive filamentous objects were observed with the addition of dextran sulfate. Some particles were also found in the cytoplasm, what might suggest the loss of nuclear membrane integrity (Supplementary Fig. 2).”

More details are presented in the new Supplementary Figure 2.

3. It is a pity that the en-block staining with uranyl acetate only does not enable to visualise the target chromatin structure in relation to large-scale chromatin organization as to see in Fig 1. Again, stronger or chromatin-specific staining could help strengthening the work.

We appreciate this comment by the reviewer, and understand that ideally if stronger uranyl acetate could provide a framework of all chromatin. However, we found out that stronger non-specific chromatin staining could hamper reliable discrimination between the target signals and unspecific ones. Therefore, we optimized the balance that was enough to demarcate the outline of the nucleus, and to reliably to distinguish the target chromatin signals from the unspecific background.

4. The Discussion should be enlarged by the biological aspect of obtained results, at least. There is only one sentence talking about the fact that the variations in the 3D chromatin folding may be a reflection of transcription activity, epigenetic state and cell cycle phase. To see transcription or even replication activity on labelled chromatin structures, it suggests itself to perform an effect experiment with incorporation of labelled uridine and/or deoxyuridine followed by immunolocalization with antibodies coupled with fluorochromes or colloidal gold particles. But I understand that this probably it is beyond the work scope.

We appreciate the reviewer's comments and suggestions for further functional characterization. We agree with the reviewer that it would be extremely interesting to see the transcriptional or replication activity in relation to the dynamics of chromatin folding, together with our method. Indeed, as the reviewer also pointed out, such interesting questions are beyond the scope of the current study. For sure, in our future directions, correlating chromatin structure and function is one of our key priorities. For instance, we could use quantum dots as additional labeling in our experimental system. Such a labeling of other molecules (RNA, DNA, or protein) could be detected for example by electron spectroscopy imaging (based on different elemental content), thus to achieve the simultaneous detection of chromatin folding, gene transcription, and protein properties. Inspired by the reviewer's suggestion, we extended our discussion in the revised manuscript in regarding the heterogeneity of chromatin folding structures that we uncovered by 3D-EMISH, and speculated possible structure-function correlation between chromatin folding dynamics and gene expression in single cells.

In the discussion section we added in page 11: "The variations in the 3D chromatin folding structures within the 229 images may be a reflection of spatiotemporal dynamics and potential functional properties such as transcription activity and epigenetic state¹⁵, and also cell cycle phase^{25,30}. Heterogeneity in the chromatin organization is observed at different levels. For example, we have shown different conformations of chromosome 1 even within the same nucleus¹⁰. There are also results obtained in human cells showing high variability in the physical distances between selected genomic loci³¹ and variability in the organization of chromatin domains¹⁶. Moreover, studies of transcription reveal expression heterogeneity³². Also in the same cells that we studied – GM12878, heterogeneity was shown using single cell sequencing approaches – scATAC-Seq and scRNA-Seq when compared to bulk studies. It suggests that variability observed by us and others could interplay with transcriptional activity. Nevertheless, the relation between dynamic changes of chromatin structure and stochasticity in gene expression is not yet fully understood^{33,34}."

5. Concerning non-specific background, how big of a part of the overall signal was represented by non-specific background? Could the authors provide one image without any image filtering?

We thank the reviewer for this question. We calculated that the volume ratio of the specific vs. non-specific signal in the analyzed regions of interest was 0.78 ± 0.40 , the non-specific signal occupied $0.0188 (\pm 0.0046)$ of the total background volume (\pm STD, 10 different ROIs). Examples of the images before and after background filtering are presented in the new Supplementary, Fig. 3 in the revised manuscript. We added a relevant text to the main manuscript and to the methods section.

Reviewer 2

The authors describe the use of serial block face scanning EM to visualize chromatin folding structure. They applied their method to image and visualize a specific chromatin folding location in the human genome and furthermore analyzed more than 200 different chromatin folding structures of human lymphoblastoid cells.

The authors describe a method that can on the one hand deliver the resolution that is needed for visualizing the fine structure of chromatin and which on the other hand is not limited in imaging depth, so that whole chromatin folding structures can be acquired. They place their work therefore between super-resolution fluorescence imaging (STORM, achievable resolution $20 \times 20 \times 50$ nm, limited in imaging depth, sequence-specific) and EM tomography ($1 \times 1 \times 1$ nm, but limited imaging depth and sequence-unspecific). The method of the authors is capable of achieving a resolution of $5 \times 5 \times 30$ nm, is practically unlimited in imaging depth and sequence-specific.

The paper is well structured and written in a very clear way, and I appreciate the effort with which figures and supplementary videos were prepared. The introduction nicely describes the state of the art and why the described work is needed and how it advances chromatin visualization.

We thank the reviewer for the positive note on our manuscript, and appreciate valuable comments.

Here some questions and points which I'd recommend to address:

1. Mention explicitly that this is the first time serial block face scanning electron microscopy is used for imaging and visualizing chromatin structure.

We thank the reviewer for the suggestion. In the revised manuscript, we now stated explicitly the following sentence in the introduction (page 4): “To our knowledge, this is the first time when SBF-SEM is applied for imaging specific chromatin folding structures.”

2. Mention in the introduction that EM tomography can achieve a resolution of 1 x 1 x 1 nm. In my opinion, this is really important.

In agreement with the reviewer's comment, we added the following statement in the introduction section in the revised manuscript (page 3-4): “The most recent effort in using EM for imaging chromatin structure is EM tomography with even higher resolution (1x1x1 nm)²⁴, in which photo-oxidized label is used to mark all DNA and to visualize the overall chromatin structures for 3D imaging in the nucleus²⁵, but within a limited depth at z-axis (250nm)” Our approach overruns the latter problems, i.e. the lack of sequence specificity and low depth of imaging in the z-dimension.

The authors mention a couple of times that they are imaging chromatin with an “unprecedented resolution” (l. 24 and 75) and only later in the discussion the reader learns that previous work achieved a much higher resolution. So I'd recommend to omit the “unprecedented resolution”, mention the 1 x 1 x 1 nm resolution of EM tomography in the introduction, so that it is from the beginning on clear that it is not only about resolution, and highlight more the other advantages of 3D-EMISH like its sequence specificity.

We agree with the reviewer's suggestion. We removed the term “unprecedented resolution” from the revised manuscript and added the information about the EM tomography resolution (see our response above). In discussion, we also highlight the superiority of 3D-EMISH over the super-resolution microscopy and the complementary advantages of our 3D-EMISH to EM tomography in imaging chromatin folding structures.

3. The authors show chromatin images acquired with iPALM to compare their method with super-resolution microscopy. It would be great (if feasible) to also see images acquired with EM tomography for comparison.

We very much appreciate the reviewer's suggestion. We fully intend to apply EM tomography as a similar strategy as 3D-EMISH for imaging of chromatin folding structures. We have started a collaboration with a laboratory that is equipped to perform EM tomography and is interested in chromatin structure/biology. Excitingly, our pilot experiments have yielded some promising preliminary results, using EM tomography in the same sample as was used for SBF-SEM-based 3D-EMISH. Briefly, as demonstrated in the images below, the 3D volume of the specimen visualized previously by EMISH of the specific chromatin target, was successfully resolved by EM tomography, and achieved the resolution at 1.3 x 1.3 x 1.3 nm. We are now in progress of further improving the experimental procedure, generating more imaging data, and developing the accompanying computational algorithms to analyze the data comprehensively, and intend to increase the sample thickness (z-stack) at least to 750 nm. Such a thickness we are going to approach using the scanning-transmission electron microscopy (STEM) modality.

We envision combining SBF-SEM-based 3D-EMISH and EM tomography which could result in a new powerful method (tentatively called EM-TISH) for imaging chromatin folding structures with an “unprecedented resolution.” We, therefore, plan to report this new method and results in a separate publication. We sincerely hope that the reviewer will understand our position.

4. From the supplementary movies it is not obvious that the axial resolution is 6 times worse than the lateral resolution. Did the authors process the data in any way in this regard?

We thank the reviewer for this question. Indeed, we performed a reslicing in the axial direction of the original image in order to represent data by cubical voxels, and so to have the same scale in all directions in the refined image. That is why the axial resolution appeared not too different from the lateral resolution. The reslicing was performed by upsampling with linear interpolation between adjacent planes. We also applied a gaussian filter of size 1 pixel (i.e. 5 nm) in the x-y planes in order to remove possible pixel noise. The isotropic scale was required by the plugin used to produce the movies (with unequal scales the structures appear in the movies unnaturally flattened), and also it was needed by the algorithm for morphological domains separation, which operates on a cubical grid. The upsampling is not obviously visible unless we rotate the structure through an angle perpendicular to the axial direction, in the supplementary movies the structures were rotated through an axis perpendicular to its longest (principle) axis, which in general is randomly oriented with respect to the slicing direction

5. The provided Github link is not working. I assume that the repository is still private and will be made public upon publication, but still wanted to mention this.

We appreciate the reviewer’s understanding. Now, the Github repository is publicly available for accessing by anyone.

Reviewers' Comments:

Reviewer #1:

Remarks to the Author:

I am satisfied with the responses and revised manuscript, however, with some reservations. I do not agree with the statement "negligible changes or deteriorations in chromatin structure" in the authors response. See the changes in the chromatin structure as well as the anti-DNA labeling pattern between supplementary figures 2b and 2c. Fortunately, the manuscript contains more truthful statements "moderate changes" or "no serious artifacts in chromatin structure". Moreover, I am not sure that the magenta arrows indicate condensed chromatin (relating mainly to left arrows in Figs. 2d and 2e and one magenta arrow in Fig 2f), because, it is almost impossible to distinguish chromatin structures from ribonucleoproteins structures without specific labeling or contrasting. There is "magenta arrows indicate condensed chromatin" in the figure legend. It should be more precisely written "arrows potentially indicate".

Reviewer #2:

Remarks to the Author:

I would like to thank the authors for thoroughly addressing all the raised points, especially for sharing their newest images with the reviewers, which are not even part of this manuscript. This looks very promising.

POINT BY POINT RESPONSE TO REFEREES' COMMENTS

Manuscript ID: [NCOMMS-19-33009-T]

We thank the reviewers for their encouraging comments and insightful suggestions. We really appreciate their time and efforts. Please find our point-by-point responses to the reviewers' comments below. For clarity, the reviewers' comments are in *italic blue* while our responses are in black.

Reviewer 1

I am satisfied with the responses and revised manuscript, however, with some reservations. I do not agree with the statement “negligible changes or deteriorations in chromatin structure” in the authors response. See the changes in the chromatin structure as well as the anti-DNA labeling pattern between supplementary figures 2b and 2c. Fortunately, the manuscript contains more truthful statements “moderate changes” or “no serious artifacts in chromatin structure”.

Moreover, I am not sure that the magenta arrows indicate condensed chromatin (relating mainly to left arrows in Figs. 2d and 2e and one magenta arrow in Fig 2f), because, it is almost impossible to distinguish chromatin structures from ribonucleoproteins structures without specific labeling or contrasting. There is “magenta arrows indicate condensed chromatin” in the figure legend. It should be more precisely written “arrows potentially indicate”.

We thank the reviewer for the positive note on our manuscript, and appreciate valuable suggestions. In agreement with her/his comments, we moderated the statements regarding the changes in the chromatin structure. Specifically, we have replaced the statement that was still present in the manuscript “*negligible changes or deteriorations in chromatin structure*” with the revised statement “**at most, moderate changes or deteriorations in chromatin structure**”. We agree with the lack of clarity in our terms regarding the electron-dense nuclear material, and changed the description of the magenta arrows as suggested by the reviewer. In the figure legend we changed the text to “**magenta arrows potentially indicate the condensed chromatin (however that cannot be uniquely distinguished from ribonucleoproteins structures without further specific labelling)**”.

Reviewer 2

I would like to thank the authors for thoroughly addressing all the raised points, especially for sharing their newest images with the reviewers, which are not even part of this manuscript. This looks very promising.

We thank the reviewer very much for the positive and encouraging notes on our efforts to improve the manuscript.

All other changes to the manuscript have been done according to the editorial requirements.